# 2′,3′ cyclic nucleotide 3′ phosphodiesterase 1 functional isoform antagonizes HIV-1 particle assembly

Shuntao Liang[1,2,*], Qin Zhang[1,*], Fang Wang[1], Shiwei Wang[1], Guoli Li[2], Dong Jiang[2], Hui Zeng[1]

IFN-stimulated gene 2′,3′ cyclic nucleotide 3′ phosphodiesterase (CNP) comprises two isoforms: the short CNP1 and the long CNP2, featuring an additional N-terminal segment of 20 amino acids (N20aa) proposed as a mitochondrial targeting sequence. Notably, CNP1 can be produced by cleaving the N20aa segment from CNP2. Although previous investigations have recognized the HIV-1 particle assembly impairment capability of CNP2, the antiviral activity of CNP1 remains ambiguous. Our study clarifies that CNP1, as opposed to CNP2, serves as the primary isoform exerting anti-HIV-1 activity. Both CNP1 and CNP2 can localize to the cell membrane, but the N20aa segment of CNP2 impedes CNP2–HIV-1 Gag interaction. Cleavage of the N20aa segment from CNP2 results in the formation of a functional, truncated form known as CNP1. Intriguingly, this posttranslational processing of CNP2 N20aa occurs within the cytoplasmic matrix rather than the mitochondria. Regulated by CTII motif prenylation, CNP1 proteins translocate to the cell membrane and engage with HIV-1 Gag. In conclusion, our findings underscore the pivotal role of posttranslational modification in governing the inhibitory potential of CNP in HIV-1 replication.

## Introduction

The IFN system constitutes the primary defense mechanism against viral infections in vertebrates. Type I IFNs initiate a signal cascade through their corresponding receptors (IFNAR), leading to the up-regulation of IFN-stimulated genes (ISGs). These genes subsequently interfere with viral replication and dissemination (Borden et al, 2007; Seo et al, 2011). Some ISGs exhibit multiple isoforms with varying degrees of antiviral activity, arising from alternative transcription, splicing variants, alternative translation or posttranslational modifications (Everitt et al, 2012; Zhang et al, 2013; Liu et al, 2016; Chen et al, 2017; Wu et al, 2017).

2′, 3′ cyclic nucleotide 3′ phosphodiesterase (CNP) stands out as a crucial ISG capable of suppressing the viral replication of HIV-1, simian immunodeficiency virus, and severe acute respiratory syndrome coronavirus 2 (SARS-CoV-2). Identified as two isoforms, CNP1 (~46 kD)

and CNP2 (~48 kD) are both encoded by a single gene with alternative transcription initiation from distinct promoters. Whereas CNP1 mRNA is predominantly expressed in the central nervous system (Staugaitis et al, 1990), CNP2 mRNA represents the major transcripts in various noncentral nervous system tissues (McFerran & Burgoyne, 1997). CNP1 proteins, translated from CNP1 mRNA or alternatively from the second start codon of CNP2 mRNA (O'Neill et al, 1997), predominantly localize on the plasma membrane (Braun et al, 1991). The C-terminal CTII motif prenylation regulates the plasma membrane localization of CNP1 (Braun et al, 1991; Cox et al, 1994; Bifulco et al, 2002). The long isoform CNP2 proteins possess an additional N-terminal 20 amino acids (N20aa) proposed as a mitochondrial targeting sequence (MTS) (Lee et al, 2006). This MTS facilitates the translocation of CNP2 proteins into the mitochondria. Cleavage of the MTS generates a truncated form identical to CNP1 proteins (Lee et al, 2006).

In a study employing transient transfection of a CNP2 cDNA plasmid into 293T cells, Wilson and colleagues reported that HIV-1 Gag proteins likely recruit CNP proteins to budding sites on the plasma membrane, thereby impeding the assembly of HIV-1 particles (Wilson et al, 2012). They demonstrated that the antiviral effect of CNP proteins is associated with the prenylation of the C-terminal domain and the presence of aspartic acid at codon 72 within the P-loop (Wilson et al, 2012).

Considering the distinction between CNP1 and CNP2 proteins, our investigation delves into whether both CNP1 and CNP2 constitute functional forms with anti-HIV-1 capacity using pseudovirus systems and doxycycline (dox) inducible expression systems. We found that (1) CNP1 is the functional isoform with anti-HIV-1 capacity; (2) posttranslational processing of CNP2 is the primary source of intracellular CNP1 protein; (3) CNP1 proteins interact with HIV-1 Gag protein on the cell membrane; (4) N20aa inhibits CNP2 interaction with Gag protein on the cell membrane.

## Results

### Truncated CNP1 proteins demonstrate anti-HIV-1 activity

To quantify the abundance of CNP variants in monocytes and CD4[+] T cells, the primary targets of HIV-1, we amplified mRNA variants of

---

[1]Biomedical Innovation Center, Beijing Shijitan Hospital, Capital Medical University, Beijing, China    [2]Institute of Infectious Diseases, Beijing Ditan Hospital, Capital Medical University, Beijing, China

Correspondence: zenghui@ccmu.edu.cn
*Shuntao Liang and Qin Zhang shared first authorship

CNPs using designed upstream primers located within their respective exon 1 and a common downstream primer situated in exon 2 (Fig S1). In resting human monocytes and CD4$^+$ T cells, the mRNA levels of the CNP2 variant were ~200 times higher than those of CNP1. After treatment with IFN-$\alpha$ and pseudotyped HIV-1 particles, both CNP1 and CNP2 transcript levels increased in these cells, with CNP2 mRNA levels remained 40–90 times higher than those of CNP1 (Fig 1A and B). Despite the higher levels of CNP2 mRNA, Western blot analysis revealed that CNP1 was the predominant CNP protein isoform in these cells, irrespective of IFN-$\alpha$ or pseudotyped HIV-1 particle treatment (Fig 1C).

To assess the antiviral potency of both CNP isoforms, we transfected 293T cells with CNP1 or CNP2 mammalian expression plasmids along with HIV-1-based lentiviral pseudotyped particle assemble vectors. Subsequently, we evaluated the levels of viral particulate capsid (p24) in purified supernatants, and the intracellular levels of CNP, p24, and Gag precursor p55 in transfected 293T cells using Western blot. Simultaneously, the infectious virion yield was monitored by luciferase activity in 293T cells at 48 h postinfection. As depicted in Fig 1E, luciferase assays demonstrated that both CNP1 and CNP2 plasmids resulted in more than a 1,000-fold reduction in luciferase activity. Both CNP1 and CNP2 plasmids significantly reduced p24 levels released in the culture media, accompanied by an increase in intracellular levels of p55 gag precursors in transfected 293T cells (Fig 1D). These findings suggest that the short protein isoform CNP1 can inhibit HIV-1 virus assembly.

## Posttranslational processing of CNP2 N20aa is prerequisite for antiviral capacity

CNP1 proteins could be translated from an alternative start site in CNP2 mRNA or generated by cleaving N20aa of CNP2 proteins (O'Neill et al, 1997; Lee et al, 2006). We further investigated the effects of translation and posttranslational processing on the antiviral activity of CNPs. To selectively impede the alternative translation of CNP1 proteins from CNP2 mRNA, we generated a CNP2-M21L mutant (Fig S2A). Compared with the CNP2 construct, the CNP2-M21L mutant reduced the expression levels of CNP1 protein and impeded the ability to block HIV-1 particle assembly (Fig 1F and G).

Next, we explored whether posttranslational processing of CNP2 proteins was crucial for antiviral activity. To inhibit the cleavage of N20aa, we constructed a Flag-CNP2 construct in which a Flag tag sequence was added to the N-terminal of CNP2 cDNA, and similarly, a Flag-CNP1 construct. We transfected the plasmids encoding CNP1, CNP2, Flag-CNP1 or Flag-CNP2 cDNA into 293T cells along with HIV-1 pseudotype particle assemble vectors. The Flag-CNP2 plasmids transfected into 293T cells exhibited reduced levels of CNP1 proteins and increased levels of CNP2 (Fig 1D), suggesting that the process forming truncated CNP1 from CNP2 was impaired. Meanwhile, in comparison with CNP2 constructs, Flag-CNP2 constructs exhibited reduced antiviral capacity (Fig 1D and E). As a control, Flag-CNP1 and CNP1 constructs showed comparable antiviral capacity (Fig 1D and E). Therefore, the posttranslational processing of CNP2 N20aa to form CNP1 is prerequisite for anti-HIV-1 activity.

## Posttranslational processing of CNP2 N20aa occurs in the cytoplasmic matrix

A previous study posited that the N20aa of CNP2 served as a MTS, regulating CNP2 translocation and protein processing within the mitochondria. Consequently, the mitochondrial targeting feature was deemed necessary for the antiviral activity of CNP. To scrutinize this hypothesis, we established stably transduced 293T cell lines expressing CNP1 and CNP2 based on the pLVX-TetOne lentiviral vector (Fig 2A). In both CNP1 and CNP2 stably transduced 293T cells, Western blot analysis revealed the presence of CNP1 proteins in the mitochondrial fractions (Fig 2B). Confocal imaging further demonstrated the co-localization of CNP1 with the mitochondrial marker Tom20 (Fig 2C). To explore the intrinsic property of mitochondrial targeting in CNP1, we deleted 30 aa and 40 aa from the N-terminal of CNP2 (Δ30 CNP2 and Δ40 CNP2, respectively) (Figs 2A and S2A). These mutant proteins also translocated into the mitochondria (Fig 2D), suggesting that CNP1 inherently possesses mitochondrial targeting properties. Therefore, we questioned the assumption that N20aa of CNP2 functioned as MTS and was cleaved in the mitochondria.

It has been reported that phosphorylation at amino acids Ser9 and Ser22 hindered the cleavage process of CNP2 and its targeting to mitochondria. Therefore, we generated stably transduced 293T cell lines with a CNP2-S9/22A mutant construct in which phosphorylation of Ser9 and Ser22 was blocked, or with a CNP2-S9/22D mutant to mimic CNP2 phosphorylation at Ser9 and Ser22 (Fig S2A). Contrary to the previous report, Western blot analysis revealed the presence of CNP2 and CNP1 proteins in the mitochondrial fraction in 293T cells stably transduced with these mutants (Fig S2B). Confocal imaging also showed the colocalization of CNP signals with Tom20 signals (Fig S2C).

Next, we sought an alternative approach to assess whether the N20aa of CNP2 functioned as an MTS. To achieve this, we engineered CNP2$^{1-20}$-GFP, CNP2$^{1-30}$-GFP, and CNP2$^{1-40}$-GFP mutants encoding fusion proteins of N-terminal 20, 30 or 40 amino acids of CNP2 fused with GFP as previous research (Fig S2D) (Backes et al, 2018). We constructed a COX8A-MTS-GFP mutant encoding a fusion protein of human COX8A MTS and GFP as a positive control (Figs S2D and 2E) (Van Kuilenburg et al, 1988). Western blot analysis and confocal imaging confirmed that COX8A MTS directed GFP translocation into the mitochondria (Fig 2F and G). In contrast, GFP signals from CNP2$^{1-20}$-GFP, CNP2$^{1-30}$-GFP, and CNP2$^{1-40}$-GFP were found in the cytosolic fraction but not in the mitochondrial fractions (Fig 2F). Confocal microscopy also revealed that GFP signals did not colocate with the mitochondria marker in CNP2$^{1-20}$-GFP, CNP2$^{1-30}$-GFP or CNP2$^{1-40}$-GFP stably transduced 293T cells (Fig 2G). Therefore, the N20aa of CNP2 did not function as MTS to mediate GFP translocation into the mitochondria.

We observed that the molecular weight of CNP2$^{1-20}$-GFP was comparable with that of GFP (Fig 2E and F), GFP signals of CNP2$^{1-20}$-GFP, CNP2$^{1-30}$-GFP, and CNP2$^{1-40}$-GFP was found in the cytosolic fraction (Fig 2F), suggesting that the N20aa of CNP2$^{1-20}$-GFP, CNP2$^{1-30}$-GFP, and CNP2$^{1-40}$-GFP was cleaved outside the mitochondria. In addition, GFP signals were not observed in the endoplasmic reticulum, trans-Golgi, cis-Golgi, cytoskeleton, and lysosomes in CNP2$^{1-20}$-GFP stably transduced 293T cells (Fig S3A).

**Figure 1. CNP1 is the functional isoform with anti-HIV-1 Gag assembly.**
**(A, B)** Differential CNP mRNA expression in monocytes and CD4+ T cells before and after IFN-α (500 IU/ml) (A) or pseudotyped HIV-1 particle treatment (B) for 24 and 48 h. The relative mRNA amounts before and after IFN-α treatment were semiquantified by real-time PCR. Data are presented as mean ± SD of three independent experiments. A two-tailed unpaired *t* test was employed to assess differences using GraphPad Prism 9. **(C)** Examination of CNP1 and CNP2 protein levels before and after IFN-α or pseudotyped HIV-1 particle treatment in monocytes and CD4+ T cells using Western blot. CNP expression after transfection of CNP2-S9/22D expression vectors in 293T cells was performed to confirm the CNP isoform. **(D, F)** Inhibition of pseudotyped HIV-1 particle assembly by CNP2, CNP1, FLAG-CNP2, FLAG-CNP1, and CNP2 M21L. Transfections of pcDNA5-CNP2,

When we used colchicine to block the assembly of microtubule proteins or Brefeldin A to disrupt the transport of proteins from the endoplasmic reticulum to the Golgi apparatus, protein processing of CNP2 and CNP2[1–20] GFP was not affected by colchicine or Brefeldin A (Fig S3B and C). These data suggest that the assembly of cytoskeletal components and the transport of proteins from the endoplasmic reticulum to the Golgi apparatus are not involved in the processing N20aa of CNP2. Moreover, by preventing the cleavage of N20aa through the addition of the Flag tag, Flag-CNP2 retains localization to the cell membrane and mitochondria (Fig S3D and E), and no localization to the endoplasmic reticulum (Fig S3F). Taken together, posttranslational processing of CNP2 N20aa might occur in the cytoplasmic matrix.

**CNP1 interacts with HIV-1 Gag proteins on the cell membrane**

Previous reports indicate that CNP blocks HIV-1 particle assembly through Gag-CNP interaction at budding sites on the plasma membrane (Wilson et al, 2012). To further investigate CNP1–Gag interaction, we generated 293T cell lines with stable expression of HIV-1 Gag proteins and dox-induced expression of CNP variants (Figs 3A and S4A). High-resolution 3D reconstruction confocal imaging confirmed the distribution of CNP1 protein on the cell membrane and in the mitochondria in the absence of HIV-1 Gag (Fig 3B and Videos 1 and 2). In the absence of CNP expression, HIV-1 Gag colocalized with the cell membrane protein CD81 rather than with the mitochondria protein Tom 20 (Fig 3C and Videos 3 and 4). Upon induction of CNP1 expression by dox, CNP1 protein co-localized with Gag protein on the cell membrane protein CD81 (Fig 3D and E, Videos 5 and 6), but not with Tom 20 (Fig S4B). Subsequently, we constructed a CNP2 D72E mutant to abolished CNP binding to Gag (Fig S4A) (Wilson et al, 2012). This mutant localized to the cell membrane (Fig S4D), but lost its anti-lentiviral assembling potency (Fig 3F and G). Thus, the primary interaction between CNP1 and HIV-1 Gag occurs on the cell membrane.

To further confirm this notion, we engineered a CNP2-C418A mutant (CNP2-CAAXM) to interfere with prenylation modifications of the CAAX box at the C-terminus and subsequently impede CNP1 protein translocation to the plasma membrane (Fig S4A). Consistent with the previous studies, Western blot showed that CNP1 proteins were enriched in the membrane fractions, whereas CNP2-C418A mutants were mainly in the cytosolic fractions (Fig S4E). High-resolution 3D reconstruction confocal imaging showed that the C418A mutant remained in the mitochondria (Fig 3H) and failed to co-localize with Gag protein on the cell membrane (Figs 3I and S4C, Videos 7 and 8). Consistent with these alterations, CNP2-C418A mutant showed reduced anti-lentiviral assembling potency (Fig 3F and G).

**N20aa may hinder the interaction between CNP2 and HIV-1 Gag**

We observed that cleavage N20aa of FLAG-CNP2 protein was impaired by a Flag tag (Fig 1D). However, Flag-CNP2 protein still localized on the cell membrane (Fig S3D) but displayed a reduced anti-lentiviral assembling potency (Fig 1D and E). We speculated that N20aa segment may hinder the interaction between CNP2 and Gag. To test this hypothesis, we employed molecular docking calculations using the MOE software to analyze the binding interactions between CNP and Gag proteins. Protein-protein docking analysis revealed a higher binding energy for CNP2-Gag docking compared with CNP1-Gag (Fig 4A), and N20aa segment increased the spatial distance between of CNP2 protein and Gag protein (Fig 4B).

To validate the docking calculations of CNP2-Gag protein interactions, we performed a fluorescence complementation assay in 293T cells as described previously (Kamiyama et al, 2016; Zhang et al, 2020). We co-expressed split super-folder GFP11-CNP1 (sfGFP11-CNP1) with Gag containing a split super-folder GFP (1–10) tag (Gag-sfGFP10) (Figs 4C and S5A). Western blot assay demonstrated that sfGFP11 at CNP2 N-terminus prevents the cleavage of the N20aa (Fig S5B). Confocal microscopy assay confirmed that sfGFP11-CNP2, sfGFP11-CNP1 and sfGFP11-CNP1-D72E still localized on the cell membrane (Fig S5C). The GFP levels in sfGFP11-CNP2 decreased by ~30% compared with sfGFP11-CNP1 and were comparable with sfGFP11-CNP1-D72E (Fig 4D and E). These results indicate that N20aa hinders the interaction between CNP2 and Gag proteins on the cell membrane.

# Discussion

The role of CNP in inhibiting HIV-1 particle assembly through its interaction with Gag has been elucidated in our study. Specifically, we have demonstrated how posttranslational modification influences the antiviral efficacy of CNP. The N20aa segment of CNP2 impedes interactions between CNP2 and HIV-1 Gag at the cell membrane. Cleavage of N20aa in CNP2 results in the generation of a functionally truncated form known as CNP1. Governed by the prenylation of the CTII motif, CNP1 proteins translocate to the cell membrane and engage with HIV-1 Gag.

Our investigation has unveiled a crucial discovery concerning the antiviral activity of both full-length and truncated CNP forms. We confirm that CNP1, as the functional isoform, possesses the capability to hinder HIV-1 Gag assembly. Cleavage of N20aa transforms CNP2 from an inactive state to an active form (Δ20 CNP2, CNP1). Disruption of the posttranslational modification, as observed in the Flag-CNP2 and CNP2-M21L mutants, diminishes the antiviral potential of CNP2. Notably, despite the higher levels of CNP2 variant mRNA in resting human monocytes and CD4[+] T cells, CNP1 emerges as the predominant CNP protein isoform in these cells, irrespective of treatment with IFN-α or exposure to pseudotyped HIV-1 particles. This phenomenon can be attributed to two factors. First, CNP1 proteins maybe translated from CNP1 mRNA or alternatively from the second start codon of CNP2 mRNA. Second, CNP1 proteins can

pcDNA5-CNP1, pcDNA5-FLAG-CNP2, pcDNA5-FLAG-CNP1, and pcDNA5-CNP2 M21L expression vectors into 293T cells were conducted with pseudotyped HIV-1 particle assembly vectors (pLenti6-Luc, pMDLg/pRRE, pREV, and pLP/VSVG). After 48 h, p24 in the viral particle supernate, CNP, p55 gag precursor, and p24 in transfected 293T cells were evaluated by Western blot. **(E, G)** Luciferase assay of pseudotyped HIV-1 production. Pseudotyped HIV-1 production was assessed by luciferase assay after the transduction of 293T cells with the supernatant for 48 h. Data are presented as mean ± SD of three independent transfection experiments. Source data are available for this figure.

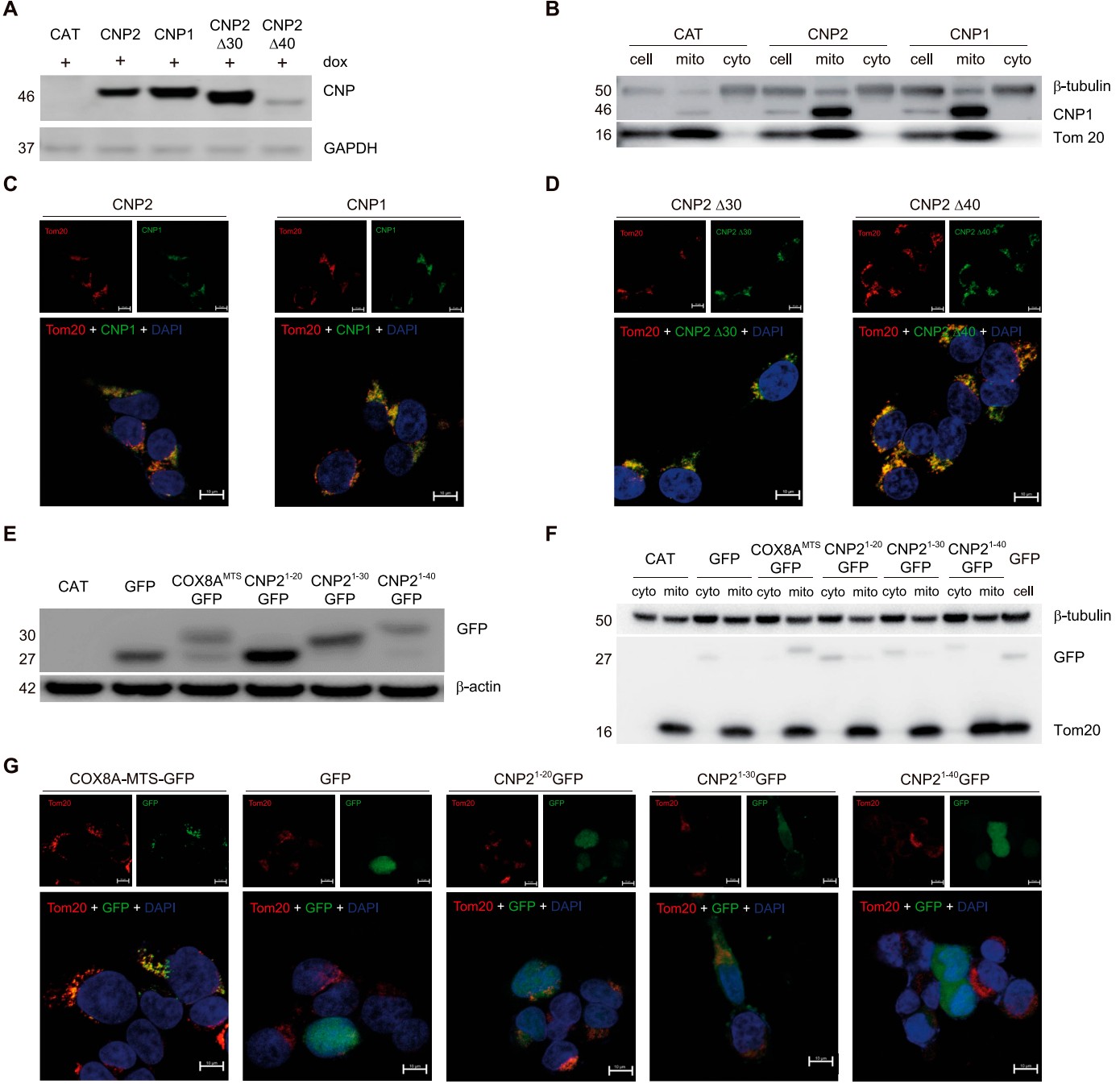

**Figure 2. N-terminal 20 amino acids of CNP2 are not mitochondrial targeting sequence.**
**(A)** 293T cell lines with doxycycline-inducible expression of WT or mutant CNP systems were cultured in medium with 50 ng/ml dox for 12 h. Western blot analysis was used to determine CNP expression, with GAPDH serving as a loading control. **(B)** CNP distribution in mitochondria and cytosolic fraction in a dox-inducible expression system after 12 h of dox induction. Differential centrifugation was employed to separate mitochondrial and cytosolic fractions, and CNP was assessed using Western blot. Tom 20 served as the mitochondria marker, and β-tubulin was the cytosol marker. Mito, mitochondria; Cyto, cytosolic. **(C, D)** Confocal analysis of CNP distribution (100× oil objective). **(A)** In a doxycycline-inducible expression system similar to (A), 293T cells were seeded on slide covers 24 h in advance, and 50 ng/ml dox was induced. After 12 h, the cells were fixed and stained for CNP (green) and Tom 20 (red) using specific antibodies, and the nucleus was stained using DAPI (blue). CNP distribution was detected by confocal imaging. **(E)** A dox-inducible GFP fusion protein expression system in 293 cell lines was cultured for 12 h in a medium with 50 ng/ml dox. Western blot was used to evaluate GFP expression, with β-actin serving as a loading control. **(B, F)** GFP or GFP fusion protein distribution in mitochondria and cytosolic fractions was analyzed by Western blot in a dox-inducible expression system as in (B). **(G)** Confocal examination of GFP distribution in the dox-inducible system of 293T. The confocal image shows the intracellular distribution of GFP (green), mitochondria stained with Tom 20 (red), and the nucleus stained with DAPI (blue) (100× oil objective). Please note that COX8A–mitochondrial targeting sequence-GFP colocalizes with mitochondria, whereas GFP, CNP2$^{1–20}$-GFP, CNP2$^{1–30}$-GFP, and CNP2$^{1–40}$-GFP did not present in the mitochondria.
Source data are available for this figure.

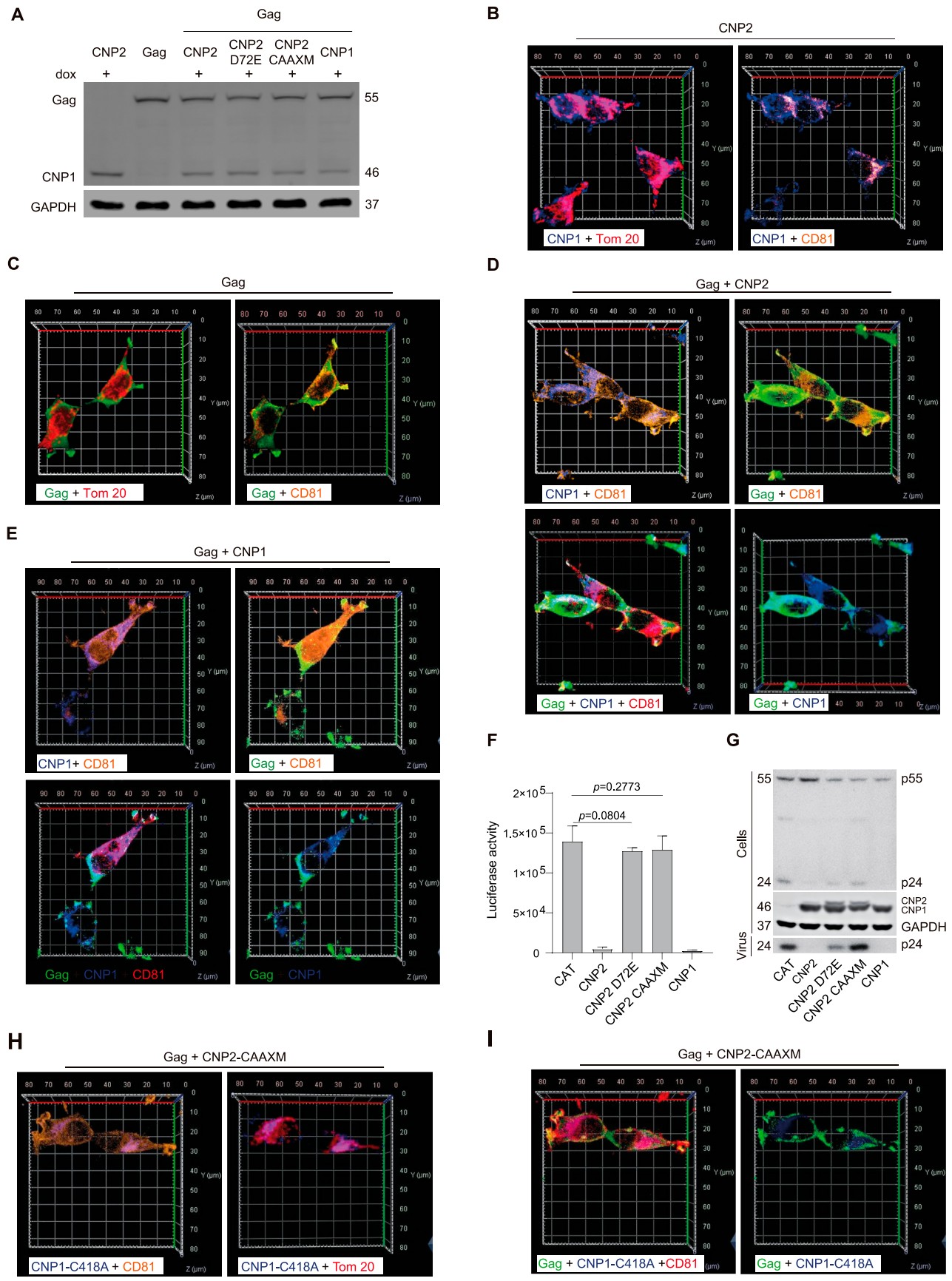

be generated from CNP2 proteins by cleaving N20aa. This dual process not only increases the quantity of CNP1 protein but also decreases the amount of CNP2 protein, thereby enhancing the host's antiviral capacity.

Similarly, analogous to CNP isoforms, human IFITM isoforms exhibit distinct roles in viral replication. For instance, Δ20 IFITM2 isoform, lacking 20aa at N-terminus of IFITM2, can be translated from full-length IFITM2 mRNA transcript (ENST00000602569, a variant of multiple transcriptional start sites), or from Δ20 IFITM2 mRNA transcript (ENST00000533141, an alternatively splice variant). In comparison with full-length IFITM2 proteins, Δ20 IFITM2 proteins demonstrate a heightened ability to restrict HIV-1 virus entry. Conversely, the Δ21 IFITM3, a 21aa truncation at the N-terminus translated from the splice variant ENST00000526811, loses its ability to restrict H1N1 IAV replication (Everitt et al, 2012; Zhang et al, 2015; Randolph et al, 2017). Notably, several ISGs have been identified to subvert viral replication through posttranscriptional modifications. Phosphorylation of Y20 leads to the redistribution of IFITM3 from the endosome to the plasma membrane, resulting in a shift from inhibiting infections to enhancing infection by SARS-CoV-2 and MERS-CoV (Compton et al, 2016). Glycosylation and GPI modification are essential for BST2 to inhibit the release of viral particles from SARS-CoV, HCoV-229E, and HIV-1 (Perez-Caballero et al, 2009; Wang et al, 2019). Phosphorylation of cGAS serves as a priming signal for activating the antivirus response (Yang et al, 2022). Therefore, the generation of different variants through either mRNA transcription or posttranscriptional modification emerges as a crucial regulatory mechanism for the antiviral capacity of ISGs.

The redistribution of ISGs plays a pivotal role as an antiviral strategy within host cells. For instance, Viperin, by interacting with HCMV gB and pp28, relocates from the endoplasmic reticulum to the Golgi apparatus, effectively inhibiting HCMV replication (Chin & Cresswell, 2001; Seo et al, 2011). In addition, Viperin can transfer from the endoplasmic reticulum to the cell membrane through its interaction with Gag, thus impeding the release of HIV-1 virus particles (Nasr et al, 2012). Our study employs specific mutants to influence CNP1's translocation to the cell membrane (C-terminal CTII motif prenylation mutant CNP2-CAAXM), or to abolish Gag-CNP1 interaction (CNP2-D72E mutant, Flag-CNP2, and sfGFP11-CNP2 proteins). This confirms that CNP1 hinders HIV-1 particle assembly via CNP1–Gag interaction on the cell membrane rather than in the mitochondria. In addition, contrary to previous suggestions by Wilson and colleagues, our findings indicate that CNP can localize on the cell membrane independently of HIV-1 Gag. Mutant CNP2-D72E, Flag-CNP2, and sfGFP11-CNP2 proteins, which exhibit defects in cleaving the N20aa and interacting with HIV-Gag, retain their ability to localize on the cell membrane. Therefore, the membrane localization of CNP is not dependent on Gag-mediated recruitment.

Taking into account the crucial role of posttranscriptional processing of N20aa in determining antiviral capacity, investigating the subcellular compartment where this processing occurs becomes imperative. N20aa segment has been previously proposed as an MTS (Lee et al, 2006). However, our findings challenge this concept, as we observed that CNP1, along with Δ30 CNP2 and Δ40 CNP2 mutants, could all localize to the mitochondria. These results indicate that the mitochondrial targeting capability of CNP is not dependent on N20aa. In addition, CNP2$^{1-20}$-GFP, CNP2$^{1-30}$-GFP, and CNP2$^{1-40}$-GFP failed to target mitochondria, instead primarily localizing within the cytosolic fraction (Fig 2F). It is noteworthy that N20aa was cleaved from CNP2$^{1-20}$-GFP, CNP2$^{1-30}$-GFP, and CNP2$^{1-40}$-GFP, rendering GFP an unreliable indicator of the localization of these mutants. Despite this limitation, our results counter the notion that N20 mediates CNP2 translocation and cleavage in the mitochondria. If N20aa functions as an MTS and is cleaved within the mitochondria, GFP signals should be observable in the mitochondria. Conversely, if N20aa is cleaved outside the mitochondria, it cannot facilitate CNP2 translocation. Confocal imaging experiments convincingly demonstrate that N20aa does not direct GFP signals to various cellular organelles, including mitochondria, endoplasmic reticulum, Golgi apparatus, cytoskeleton, and lysosomes. Intervention of cytoskeleton assembly and proteins transport from the endoplasmic reticulum to the Golgi apparatus did not affect processing of CNP2. Prevention the cleavage of N20aa through the addition of the Flag tag, Flag-CNP2 retains localization to the cell membrane and mitochondria. Considering this evidence, it is highly probable that the cleavage of the N20aa of CNP2 occurs within the cytomatrix. Further investigations are warranted to elucidate the molecular mechanisms governing the posttranslational processing of CNP2, with a specific emphasis on identifying the protease responsible for cleaving N20aa from CNP2.

In summary, our findings pinpoint CNP1 as the critical isoform responsible for inhibiting HIV-1 replication. Further investigations are needed to comprehend the intricacies of the processing of the N-terminal signal peptide of CNP2. The identification of these novel mechanisms holds promise for the development of pharmaceuticals targeting HIV-1 infection. Moreover, beyond its inhibitory effects on HIV-1, CNP has been shown to inhibit the replication of other RNA viruses (Martin-Sancho et al, 2021). Whether the antiviral

**Figure 3. CNP1 interacts with HIV-1 Gag on the cell membrane.**
**(A)** Establishment of an HIV-1 Gag stable expression and dox-inducible CNP expression system in 293T cell lines. The system was induced for 12 h with 50 ng/ml dox. Western blot was employed to evaluate CNP and p55 Gag precursor expression, with GAPDH as a loading control. **(B)** 3D reconstruction confocal image showing CNP1 co-localized with the cell membrane and mitochondria in CNP2 stable transduced 293T cell lines. Images were acquired by the Airyscan Fast module (100× oil objective). CNP1 (blue), Tom 20 (red), and CD81 (orange) on the same cells were immunolabeled with tyramide signal amplification (TSA), where CD81 served as the cell membrane marker. **(C)** Similar to Fig 3B, Gag (green), Tom 20 (red), and CD81 (orange) of 293T cells were labeled by TSA. Single-cell 3D reconstruction confocal image of Gag stable transduced 293T cells was acquired by the Airyscan Fast module (100× oil objective). Gag distributed on the cell membrane and not in the mitochondria. **(D, E)** Similar to Fig 3B and C, CNP1 (blue), Gag (green), and CD81 (orange) of 293T cells were labeled by TSA. Single-cell 3D reconstruction confocal image of 293T was acquired by the Airyscan Fast module (100× oil objective). CNP1 and Gag distributed on the cell membrane. **(F, G)** Co-transfection of 293T cells with pseudotyped HIV-1 particle assembly plasmids (pLenti6-Luc, pMDLg/pRRE, pREV, and pLP/VSVG) and plasmids expressing CNP2, CNP2 D72E, CNP2-CAAXM or CNP1. At 48 h post-transfection, infectious virion yield was measured by luciferase assay after transfection of 293T cells for 48 h. p24 and CNP abundance in purified supernatants, and p55 Gag precursor, p24, and CNP expression in cell lysates, were monitored by Western blot. Data are representative of three experiments. **(H, I)** CNP1-C418A (blue) did not localize on the cell membrane with Gag (green), and CNP1-C418A (blue) was distributed in the mitochondria.
Source data are available for this figure.

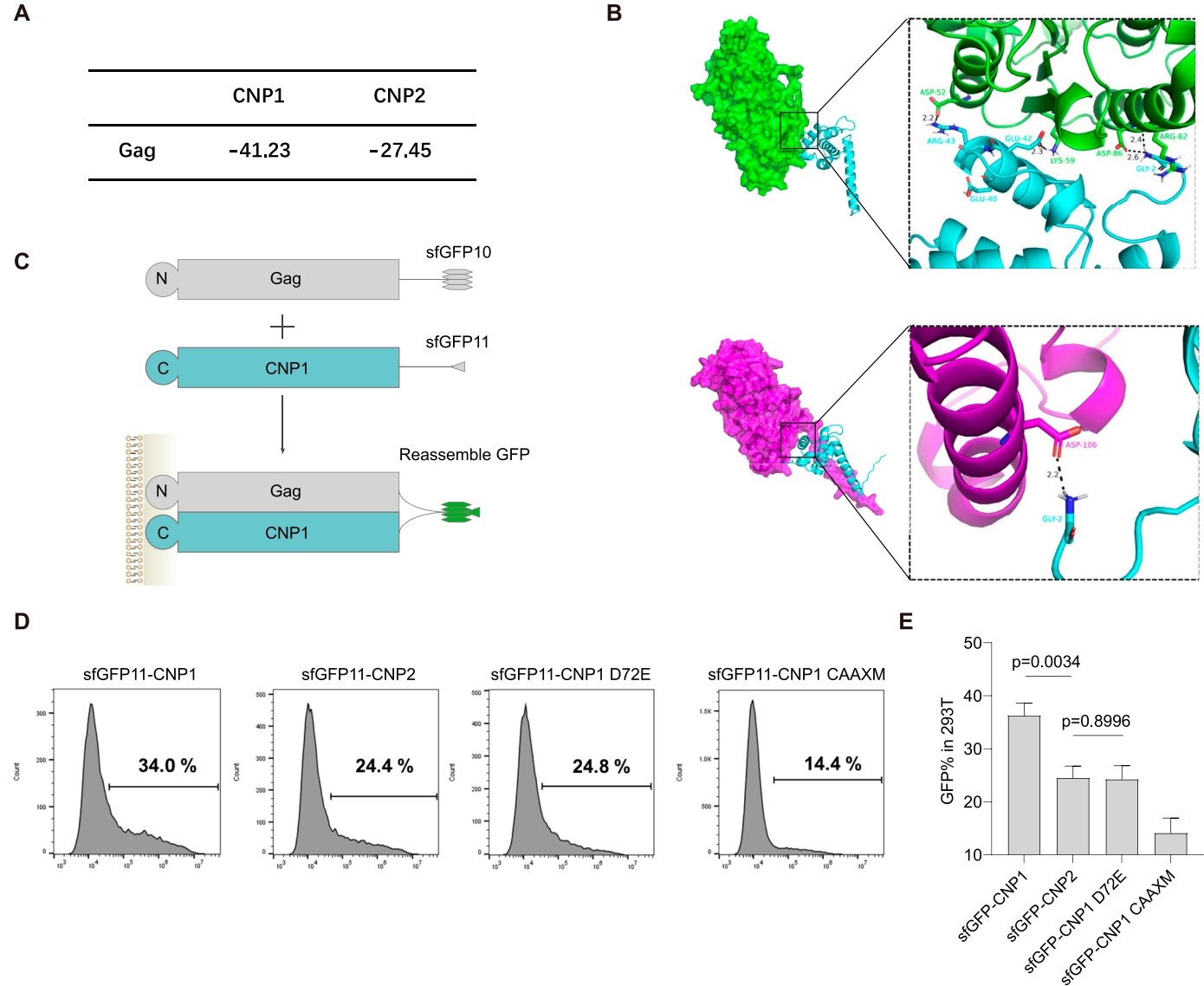

Figure 4. **N-terminal 20 amino acids of CNP2 blocks CNP2 interaction with HIV-1–Gag on the cell membrane.**
**(A)** Calculation of the binding energy for CNP2-Gag docking and CNP1-Gag docking. **(B)** Comparison between the crystallographic structures of CNP1-Gag docking and CNP2-Gag docking. Crystallographic structures of CNP1 and CNP2 were predicted by AlphaFold, whereas the crystallographic structure of Gag was obtained from PDB (ID: 1UPH). Asp52 (CNP2 D72), Lys59, Asp86, and Asp82 in the CNP1 protein (blue) formed hydrogen bonds with Arg43, Glu42, and Gly2 in the Gag protein (turquoise) respectively, with distances of 2.2, 2.3, 2.6, and 2.4 Å, respectively. Glu40 in the Gag protein also contributed to the formation of the binding pocket for the proteins. Asp106 in the CNP2 protein (aubergine) formed a hydrogen bond with Gly2 in the Gag protein, with a distance of 2.2 Å, whereas the binding between CNP2 Asp72 and Gag Glu40 disappeared. **(C)** Schematic diagram for the fluorescence complementation assay. sfGFP10: 1–10 $\beta$-strands of the GFP fragment. sfGFP11: GFP the 11th $\beta$-strand fragment. **(D, E)** Co-expression of Gag-sfGFP10 and sfGFP11-CNPs in 293T cells. FACS analysis was performed to determine the complemented GFP signal. Quantification of GFP signal was done by FlowJo (mean ± SD). *P*-values are indicated (two-tailed *t* test, n = 3).

activity of the CNP1 isoform extends to other RNA viruses universally remains a subject for future research.

# Materials and Methods

## Cell lines and cell culture

293T cells were cultured in DMEM supplemented with 10% FBS and 1% penicillin and streptomycin (PS). Human CD4$^+$ T cells and monocytes were cultured in Roswell Park Memorial Institute 1640 Medium (RPMI-1640) supplemented with 10% heat-inactivated FBS and 1% PS. All cells were gifts from members of the Beijing Key Laboratory of Emerging infectious Diseases.

## Western blot

For Western blot analysis, cells were lysed in lysis buffer (20 mM Tris–HCl, 150 mM NaCl, 1 mM EDTA, 1% TritonX-100, and protease inhibitors, pH 7.5) at 4°C for 30 min. Equal amounts of cell lysate

(25 $\mu$g) were separated on 4–12% or 10% acrylamide gels (Thermo Fisher Scientific) and transferred onto 0.22 $\mu$m PVDF membranes (Millipore). After blocking with blocking buffer (TBST supplemented with 5% nonfat dry milk), membranes were probed with specific antibodies: mouse anti-CNP (ab6319, 1:2,000; Abcam), goat anti-HIV1 p24 (ab53841, 1:2,000; Abcam), rabbit anti-Tom20 (ab186735, 1:1,000; Abcam), rabbit anti-GAPDH (5174S, 1:1,000; Cell Signaling Technology), rabbit anti-Flag (14793S, 1:1,000; Cell Signaling Technology), and rabbit anti-$\beta$-tubulin (ab6046, 1:1,000; Abcam). Subsequently, blots were probed with species-specific IRdye secondary antibodies, and visualization was performed using an Odyssey infrared imaging system (Li-COR).

### CNP transcript variants quantification by real-time RT–PCR

To quantify the relative abundance of CNP transcript variants, real-time RT–PCR was employed. Primers CNP1-F (to amplify CNP1 mRNA, GenBank accession number: NM_001330216.2): GGAGAGCTTCAGA-CAAGCTTCC, CNP2-F (to amplify CNP2 mRNA, GenBank accession number: NM_033133.4), CTCCGCGCAGGCGGGCGGCC, and commonly used downstream CNP-R, AGAGCGTCTTGCACTCTAGC were used to discriminate between different transcript variants. GAPDH-F, GAATGGGCAGCCGTTAGGAA, and GAPDH-R, GATCTCGCTCCTGGAA-GATG were used to amplify GAPDH as a control (Fig 1A). RNA and protein were isolated from healthy donor human monocytes and CD4$^+$ T cells before and after stimulation with IFN-$\alpha$ (500 IU/ml) or pseudo-typed HIV-1 particles for 24 or 48 h. The relative amount of mRNA before and after IFN-$\alpha$ treatment was semi-quantified by real-time PCR. CNP1 and CNP2 proteins before and after IFN-$\alpha$ treatment were determined by Western blot.

### Construction of CNP isoforms and mutants

The CNP2 cDNA fragment was amplified from the cDNA pool of reverse-transcribed total RNA from CD4$^+$ T cells and cloned into pcDNA5. PCR-based mutagenesis was employed to construct CNP1 and various motif mutants. CNP2-S9/22A, with phosphorylation sites at Ser9 and Ser22 mutated to non-phosphorylated Ala, and CNP2-S9/22D, with Ser replaced by phosphomimic Asp, were constructed. CNP2-M21L, where Met21 was mutated to Leu to abolish CNP1 translation by direct translation at the second ATG, was also generated. The C-terminal CAAX box of the prenylation site CTII was mutated to ATII based on CNP2 to construct CNP2-CAAXM. CNP2-D72E, with aspartic acid at position 72 involved in CNP-Gag interaction mutated to Glu, was created. C-terminal CAAX box of prenylation site CTII was mutated to ATII to construct CNP2-CAAXM.

### Antiviral potency test of different CNP mutants

The antiviral potency of these mutants was assessed by measuring pseudotyped viral particle production and subsequent luciferase assay. 293T cells were transfected with 1 $\mu$g CNP mutant constructs along with the third pseudotyped HIV-1 particle assembly system, which included four plasmids named pLenti6-Luc (1.33 $\mu$g), pMDLg/pRRE (0.67 $\mu$g), pRSV-Rev (0.5 $\mu$g), and pLP/VSVG (0.5 $\mu$g) in a six-well plate using TurboFect Transfection Reagent (Thermo Fisher Scientific). 48 h post-transfection, the virion-containing supernatant

(1,000 $\mu$l) was layered onto 400 $\mu$l of 20% sucrose in PBS, and viral particles in the supernatant were collected by centrifuging at 20,000$g$ for 120 min at 4°C. CNP, p55 Gag precursor, and p24 in the viral particle and transfected 293T cell were also tested by Western blot.

Pseudotyped HIV-1 production in the supernatant was determined by Luciferase assay. 293T cells were seeded in a 96-well black-walled plate 24 h in advance and infected with virion-containing supernatant (100 $\mu$l/well) for 4–6 h and then replaced with DMEM complete medium (200 $\mu$l/well). After 48 h of infection, luciferase activities were monitored by luminometry in a TopCounter (Perkin Elmer). A two-tailed unpaired $t$ test was used to analyze differences between CNPs and control using GraphPad Prism 9 software.

### Generation of HIV-1 Gag stable expression cell lines

The native codon Gag in pMDLg/pRRE vector was optimized and cloned into pCMV3 (pCMV3-synGag). 293T cells expressing HIV-1 Gag (293T Gag$^+$) were generated by transduction with pCMV3-synGag vectors, hygromycin selection, and maintenance of single-cell clones in DMEM supplemented with hygromycin. This cell line was routinely tested for the subcellular localization of Gag.

### Generation of doxycycline-inducible expression system in 293T cells

To establish a doxycycline-inducible expression system in 293T cells, the CNP2 WT and mutant cDNA fragments were amplified by PCR using primers derived from the aforementioned CNP cDNA in pcDNA5. Subsequently, the PCR products were cloned into the transfer plasmid pLVX-TetOne (Takara), which was digested with BamHI and EcoRI restriction enzymes. This process resulted in the creation of DNA constructs, including Dox-CNP2, Dox-CNP2 S9/22A, Dox-CNP2-S9/22D, Dox-CNP2-M21L, Dox-CNP1, Dox-CNP2 D72E, Dox-CNP2-CAAXM, and Dox-CNP2-CAAXM, using the One Step Cloning Kit (Vazyme Biotechnology). Truncated CNP mutants, named CNP2 Δ30 and CNP2 Δ40, lacking various lengths at the N-terminus, were also constructed into the pLVX-TetOne vector using the same protocol. In addition, the sequence encoding the human COX8A mitochondrial targeting signal fused with GFP protein (COX8A-MTS-GFP) was constructed into the pLVX-TetOne vector. This was achieved by amplifying the human COX8A MTS sequence, the first 25 amino acids at the N-terminus, from the cDNA pool of reverse-transcribed total RNA from CD4$^+$ T cells. This sequence was then fused to the N-terminal of the GFP encoding sequence by PCR to generate the COX8A-MTS-GFP sequence. Similarly, CNP N-terminal fusions with GFP protein, named CNP$^{1–20}$ GFP, CNP$^{1–30}$ GFP, and CNP$^{1–40}$ GFP, were also constructed into the pLVX-TetOne vector using the same protocol.

To produce lentivirus particles, 5 × 10$^5$ 293T cells were seeded into a six-well plate 24 h before transfection. Subsequently, the cells were transfected with 2 $\mu$g of the abovementioned doxycycline-inducible expressing transfer plasmids, along with 1 $\mu$g pMDLg/pRRE, 0.5 $\mu$g pRSV-Rev, and 0.5 $\mu$g pLP/VSVG, using TurboFect Transfection Reagent (Thermo Fisher Scientific). After 8 h of

transfection, the cells were replenished with 6 ml DMEM complete medium. Supernatants were harvested after 48 h of transfection. Cell-free supernatants were obtained by centrifugation for 10 min at 5,000$g$, and lentivirus particles were concentrated by centrifugation through a 20% sucrose cushion at 20,000$g$ for 120 min at 4°C. Virus particles were resuspended in DMEM medium.

To generate doxycycline-inducible expressing 293T cells, these viruses were used to infect 293T cells or Gag stable expressing 293T cells. Stable transduced cell lines were screened in DMEM complete medium complemented with 1 $\mu$g/ml puromycin (InvivoGen). To test the doxycycline-inducible expression of the proteins of interest, the cell lines were seeded into a six-well plate to adhere overnight and the DMEM medium was replaced with medium supplemented with 50 ng/ml doxycycline (Takara) for 12 h. The induced cells were then examined by Western blot using an anti-CNP antibody (Abcam) or anti-GFP antibody (Beyotime Biotechnology).

### Separation of mitochondrial and cytosolic components

Doxycycline-inducible expressing 293T cell lines were seeded into a six-well plate and supplemented with 50 ng/ml doxycycline (Takara). 12 h post-induction, mitochondrial and cytosolic fractions were separated by differential centrifugation using the Cell Mitochondria Isolation Kit (Beyotime Biotechnology). In brief, induced cells were trypsinized, washed with cold PBS, and homogenized in homogenizing buffer. The homogenate was spun at 600$g$ for 10 min, and mitochondria in the supernate were pelleted at 11,000$g$ for 10 min, and then resuspended in mitochondria storage buffer. The homogenate without mitochondria was centrifuged at 12,000$g$ for 10 min, and the supernate was labeled as the cytosolic fraction. Protein concentration was determined by the BCA Protein Assay Kit (Thermo Fisher Scientific). Both mitochondria and cytosolic fractions were subjected to Western blot. An equal relative amount of mitochondria and cytosolic protein was loaded to maintain the original mitochondria/cytosolic ratio within the transfected cell between different transfections.

### Confocal microscopy assay

Doxycycline-inducible expressing 293T cell lines were seeded on slide covers and induced with 50 ng/ml doxycycline for 12 h. The induced cells were fixed with 1% PFA in PBS for 30 min at 4°C and permeabilized with 0.1% Triton X-100 in PBS for 5 min at RT. CNP or GFP distribution was detected by immune-fluorescence staining and confocal imaging. CNP was detected using mouse anti-CNP (ab6319, 1:500; Abcam) and Alexa Fluor 488-conjugated goat anti-mouse antibody (ab150113, 1:1,000; Abcam); Mitochondrial marker Tom20 was detected using rabbit anti-Tom20 (ab186735, 1:500; Abcam) and Alexa Fluor 555-conjugated goat anti-rabbit (ab150114, 1:1,000; Abcam); nuclei were stained with DAPI (ab285390, 1:1,000; Abcam). The confocal microscopy assay was performed with the confocal laser scanning microscopy platform Zeiss LSM 880 with Airyscan under a ×100 objective, and pictures were analyzed by ZEN blue 3.2 (Zeiss) software.

### Multiplex tyramide signal amplification (TSA)

Doxycycline-inducible 293T cells expressing stable HIV-1 Gag protein were seeded on slide covers and induced with 50 ng/ml doxycycline for 12 h. After fixation with 1% PFA in PBS for 30 min at 4°C and permeabilization with 0.1% Triton X-100 in PBS for 5 min at RT, multiplex fluorescence labeling was executed using TSA-dendron-fluorophores (NEON 5-color All-round Discovery Kit for ICC, NECC550; Histova Biotechnology). Briefly, endogenous peroxidase was quenched with 3% H$_2$O$_2$ for 30 min at RT, followed by treatment with a blocking reagent for 30 min at 37°C in a humidified chamber. Primary antibodies were incubated for 90 min in a humidified chamber at 37°C, followed by washing with 0.1% Tween-20 in PBS three times. HRP-conjugated secondary antibodies were then incubated for 1 h in a humidified chamber at 37°C and subsequently washed three times with 0.1% Tween-20 in PBS. TSA visualization was performed in a reaction buffer for 30–60 s. The primary and secondary antibodies were thoroughly eliminated by Abcracker buffer for 30 min in a humidified chamber at 37°C. Each antigen was labeled with distinct fluorophores in a serial fashion. The multiplex antibody panels applied in this study included rabbit anti-CD81 (ab244297, 1:250; Abcam) and GAR-HRP (PV-6001; Zsbio) labeled by NEON420 (NECC550, 1:500; Histova Biotechnology); mouse anti-CNP (ab6319, 1:500; Abcam) and GAM-HRP (PV-6002; Zsbio) labeled by NEON670 (NECC550, 1:500; Histova Biotechnology); goat anti-HIV1 p24 (ab53841, 1:500; Abcam) and RAG-HRP (PV-9003; Zsbio) labeled by NEON520 (NECC550, 1:500; Histova Biotechnology); rabbit anti-Tom20 (ab186735, 1:500; Abcam) and GAR-HRP (PV-6001; Zsbio) labeled by NEON600 (NECC550, 1:500; Histova Biotechnology). After sequential detection of all antibodies, the cells were stained with SN470 (NECC550, 1:500; Histova Biotechnology) to label the nucleus at RT for 10 min. For 3D super-resolution cell imaging, multiplex TSA-stained cell slides were imaged using the confocal laser scanning microscopy platform Zeiss LSM 880 with Airyscan under a ×100 objective. Reconstruction and image analysis of the 3D images were performed using ZEN blue 3.2 software (Zeiss).

### Extraction of transmembrane proteins and cytosolic protein fraction

Transmembrane proteins were extracted using the ProteoExtract Transmembrane protein extraction kit (TB477; Novagen). Doxycycline-inducible CNP-expressing 293T cell lines, capable of stable expression of HIV-1 Gag protein, were seeded in six-well plates and induced with 50 ng/ml doxycycline for 12 h. A total of 0.1 ml Extraction Buffer 2A was prepared by mixing 50 $\mu$l Extraction Buffer 2 and 50 $\mu$l TM-PEK Reagent A. The 293T cells were washed twice with cold PBS, transferred to a 1.5-ml conical tube, and collected by centrifugation at 800$g$ for 5 min at 4°C. Cells were resuspended in 0.1 ml Extraction Buffer 1, and 5 $\mu$l Protease Inhibitor Cocktail Set III was added. After incubation for 10 min at 4°C with gentle agitation, the cells were collected by centrifugation at 1,000$g$ for 5 min at 4°C. The supernatant was carefully removed and referred to as the cytosolic protein fraction. The pellet was resuspended in 0.1 ml Extraction Buffer 2A, and 5 $\mu$l of Protease Inhibitor Cocktail Set III was added. After incubation for 1 h at 4°C with gentle agitation, the mixture was

centrifuged at 16,000g for 15 min at 4°C. The supernatant, enriched in integral membrane proteins, was collected.

## Supplementary Information

## Acknowledgements

This work was supported by The National Natural Science Foundation of China (31170853), The Beijing Hospital Authority (DFL20191801), and the high-level public health talents (lingjunrencai-02-06).

### Author Contributions

S Liang: data curation, formal analysis, validation, investigation, visualization, methodology, and writing—original draft, review, and editing.
Q Zhang: data curation, software, formal analysis, investigation, visualization, methodology, and writing—original draft, review, and editing.
F Wang: data curation, software, formal analysis, validation, visualization, and methodology.
S Wang: data curation, software, formal analysis, validation, and visualization.
G Li: data curation, software, investigation, visualization, and methodology.
D Jiang: conceptualization, resources, funding acquisition, and methodology.
H Zeng: conceptualization, resources, data curation, supervision, funding acquisition, validation, investigation, methodology, project administration, and writing—original draft, review, and editing.

### Conflict of Interest Statement

The authors declare that they have no conflict of interest.

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
