## [Reviewer comments · Life Science Alliance]

Life Science Alliance

2',3'Cyclic nucleotide 3'phosphodiesterase 1 functional isoform antagonizes HIV-1 particles assembly

Shuntao Liang, Qin Zhang, Fang Wang, Shiwei Wang, Guoli Li, Dong Jiang, and Hui Zeng

DOI: <https://doi.org/10.26508/lsa.202302188>

Corresponding author(s): Hui Zeng, Beijing shijitan Hospital, Capital Medical University

Review Timeline:

Submission Date:	2023-05-30
Editorial Decision:	2023-08-04
Revision Received:	2023-11-21
Editorial Decision:	2023-12-15
Revision Received:	2023-12-19
Accepted:	2023-12-21

Transaction Report:

August 4, 2023

Re: Life Science Alliance manuscript #LSA-2023-02188-T

Shuntao Liang
Capital Medical University
Beijing Ditan Hospital
China

Dear Dr. Liang,

Thank you for submitting your manuscript entitled "CNP1 is the functional isoform antagonizing HIV-1 particles assembly" to Life Science Alliance. The manuscript was assessed by expert reviewers, whose comments are appended to this letter. We invite you to submit a revised manuscript addressing the Reviewer comments.

Thank you for this interesting contribution to Life Science Alliance. We are looking forward to receiving your revised manuscript.

Sincerely,

B. MANUSCRIPT ORGANIZATION AND FORMATTING:

Reviewer #1 (Comments to the Authors (Required)):

This manuscript presents data on the synthesis and anti-HIV activities of the two isoforms of 2',3'-cyclic nucleotide 3'-phosphodiesterase. The results show that the shorter form possesses the antiviral activity and suggest that it is formed principally by N-terminal truncation of the longer form. These are significant results on an interesting topic. There are a few points that need correction or clarification. I would urge the authors to use a logarithmic scale for the data in Fig. 1E, so that the low values can still be seen by the reader. Line 136 should be expanded so that the purpose of the M21L mutant of CNP2 is clearer. In Fig. 1F and 1G, how much M21L plasmid was used ? In Fig. 3F, I believe the heading should be Gag + CNP1, not Gag + CNP2. The plasmids listed in lines 359-360 should be identified with complete specificity. Finally, the English in the manuscript is unfortunately full of mistakes. It must be completely edited by a native English speaker.

Reviewer #2 (Comments to the Authors (Required)):

In this manuscript, Liang et al. endeavour to elucidate the restriction of human immunodeficiency virus 1 (HIV-1) assembly by 2', 3'-cyclic nucleotide 3'-phosphodiesterase proteins (CNPs). CNPs are interferon stimulated genes (ISGs) that have been previously shown to suppress viral replication of HIV-1, simian immunodeficiency virus (SIV), and severe acute respiratory syndrome coronavirus 2 (SARS-CoV-2). Two isoforms of CNPs have been identified, CNP1 and CNP2. CNP1 and CNP2 are encoded by a single gene with alternative transcription initiation from respective promoters. Compared to CNP1 proteins, CNP2 proteins have an extra N-terminal 20 amino acids, which have been recognized as mitochondrial targeting sequences (MTS). While CNP2 has been previously shown to block HIV-1 particle assembly, it is unclear whether CNP1 has the same antiviral activity. The authors address this question by using a pseudovirus system and a doxycycline (dox) inducible expression system. To elucidate the localization of CNP proteins in the cell and the site of their interaction with HIV-1 Gag protein, they performed a high-resolution 3D reconstruction confocal imaging. The authors conclude that 1) CNP1, a truncated form of CNPs, has anti-HIV-1 activity, 2) post-translational processing of N-terminal 20 amino acids of CNP2 occurs in the cytomatrix and is prerequisite for its antiviral activity, 3) CNP1 proteins interact with HIV-1 Gag protein on the cell membrane, and 4) N-terminal 20 amino acids of CNP2 inhibit CNP2 interaction with Gag protein on the cell membrane. This conclusion is based on four key pieces of data:

- 1) Co-transfection of either CNP1 or CNP2 expression plasmid with HIV-1-based lentiviral pseudotyped particle assembly vectors in 293T cells results in more than 1,000-fold reduction of luciferase activity and reduced p24 levels in the culture media/increase of Pr55Gag precursors in transfected 293T cells. - Suggesting that CNP1, a truncated form of CNP2, has anti-HIV-1 activity.
- 2) To inhibit the cleavage of N-terminal 20 amino acids of CNP2, they generated Flag-CNP2. To eliminate alternative CNP1 proteins' translation from CNP2 mRNA, they generated a CNP2-M21L mutant. Co-transfection of either Flag-CNP2 or CNP2-M21L expression plasmid with HIV-1-based lentiviral pseudotyped particle assembly vectors in 293T cells exhibits reduced levels of CNP1 proteins and enhanced levels of CNP2. Compared to wild-type CNP2, Flag-CNP2 shows reduced anti-HIV-1 activity. - Suggesting that N-terminal 20 amino acids of CNP2 post-translational processing to form CNP1 is prerequisite for anti-HIV-1 activity.
- 3) Generation of 293T cells with a stable expression of HIV-1 Gag proteins and dox-induced expression of CNP variants; upon CNP1 induction by dox, CNP1 co-localized with HIV-1 Gag proteins and cell membrane protein CD81 but not with mitochondria protein Tom 20. - Suggesting that CNP1 interacts with HIV-1 Gag on the cell membrane.
- 4) Co-expression of either split super-folder GFP11-CNP1 (sfGFP11-CNP1) or sfGFP11-CNP2 with Gag containing a split sfGFP10 (Gag-sfGFP10) in 293T cells shows approximately 30% decrease in GFP level in sfGFP11-CNP2 compared to sfGFP11-CNP1. - Suggesting that the N-terminal 20 amino acids of CNP2 inhibit CNP2 interaction with Gag protein on the cell membrane.

Major Points

- 1) The abstract does not read well; in particular, lines 50-55, and needs to be rewritten.
- 2) Lines 109-111 and Fig. 1C: Observation that CNP1 is the major CNP protein isoform in monocytes and CD4+ T cells

regardless of any treatment is very interesting; however, it is never further analysed or discussed. The fact that Western blot shows different protein expression compared to the mRNA data in Fig. 1A should be discussed further, perhaps along lines 260-262 of the discussion where this topic is somehow brought up.

- 3) Lines 160-162 and Fig. S2B: Observation that CNP2 phosphorylation mutants do not show predicted localization is never addressed/discussed. Possible reasons for these results should be discussed in the discussion.
- 4) Lines 171-173 and Fig. 2F and 2G: Could GFP tagging cause protein's mislocalization in the cell? This possibility is not mentioned and should be either addressed in the results section or discussed in the discussion. Furthermore, in lines 174-176, it is mentioned that the N-terminal 20 amino acids of CNP2 were cleaved from GFP; if true, is this really a good experiment to do to determine the localization of the N-terminal 20 amino acids of CNP2? This should be discussed further and/or supported by additional experiments such as using an HA-tagged N-terminal 20 amino acids of CNP2, which can be stained with an anti-HA primary antibody followed by a secondary GFP conjugated antibody.
- 5) Lines 231-232: Is the hypothesis of this statement/conclusion that tagging CNP2 at its N-terminus with a sfGFP11 prevents the cleavage of the N-terminal 20 amino acids, as Flag tag did, and so you can observe the non-cleaved sfGFP11-CNP2 at the cell membrane? This should be explicitly stated in this results section as it is critical to understanding this results section's conclusion.
- 6) General comment about the conclusion: A lot of it is focused on other, similar ISGs and their antiviral activity, which is fair and important to include as they share some similarities with CNPs; however, the discussion should include/address multiple observations from this study, some of which I have pointed out in my comments (see Major Points 2, 3, and 4).

Minor Points

- 1) Figure 1 label, line 11: "transfected with CNP2, ..."
- 2) Line 193: The fact that post-translational processing of the N-terminal 20 amino acids of CNP2 may occur in the cytomatrix should be further elaborated on in the discussion; experiments needed to determine whether post-translational processing indeed occurs in the cytomatrix could be proposed.

Other observations: Materials and Methods section is very well written!

Response to Editor & Reviewers

We have made the necessary revisions to our paper and believe that your comments have greatly contributed to the clarity and improvement of our work. Please refer to our response (highlighted in blue) to the specific comments provided by the reviewer (presented in black) below.

According to the referees' suggestion, we have checked and revised the paper.

The change of Figure number in Manuscript and Supporting Information.

Original	Now (Revised)
Fig 1	Fig 1
Fig 2	Fig 2
Fig 3	A new Fig 3
Fig 4	Fig 4
Fig S1	Fig S1
Fig S2	Fig S2
Fig S3	A new Fig S3E
Fig S4	Fig S4
	A new Fig S5

Responses to Reviewer #1

1. Q : I would urge the authors to use a logarithmic scale for the data in Fig. 1E, so that the low values can still be seen by the reader.

A : Thank you for your suggestion regarding the data presentation. As per your recommendation, we have adjusted the y-axis a logarithmic scale in Figure 1E.

Q : Line 136 should be expanded so that the purpose of the M21L mutant of CNP2 is clearer.

A: Thank you for your suggestion. We have revised this paragraph (Line 118). We hope that the purpose of the M21L mutant of CNP2 is clear to the readers now.

Q: In Fig. 1F and 1G, how much M21L plasmid was used?

A: We used 1000ng of CNP2 M21L plasmid DNA for cell transfection. Accordingly, we have revised Figures 1F and 1G.

Q: In Fig. 3F, I believe the heading should be Gag + CNP1, not Gag + CNP2.

A: In the revised manuscript, Fig 3F has been amended to Fig 3D. The heading for Figure 3D is accurately labeled as “Gag + CNP2”. In this Figure, we employed CNP2 stable transduced and HIV-1 Gag protein stable expression 293T cell lines to investigate the distribution of CNP1 (cleavage product of CNP2). The data of “Gag + CNP1 are presented in Figure 3E.

Q: The plasmids listed in lines 359-360 should be identified with complete specificity.

A: According to your valuable suggestion, we have provided more specific identification about plasmids in the revised version (Line 361-364).

Q: Finally, the English in the manuscript is unfortunately full of mistakes. It must be completely edited by a native English speaker.

Q: Thanks very much for your comments. We have asked native English speaker to polish our paper.

Reviewer #2 (Comments to the Authors (Required)):

Major Points

1. Q: The abstract does not read well; in particular, lines 50-55, and needs to be rewritten.

A: Thank you for your valuable suggestion. We have revised the abstract.

2. Q: Lines 109-111 and Fig. 1C: Observation that CNP1 is the major CNP protein isoform in monocytes and CD4⁺ T cells regardless of any treatment is very interesting; however, it is never further analysed or discussed. The fact that Western blot shows different protein expression compared to the mRNA data in Fig. 1A should be discussed further, perhaps along lines 260-262 of the discussion where this topic is somehow brought up.

A: We have added a discussion on this topic in the manuscript (lines 241-248). The revised lines are as follows:

“Notably, despite the higher levels of CNP2 variant mRNA in resting human monocytes and CD4⁺ T cells, CNP1 emerges as the predominant CNP protein isoform in these cells, irrespective of treatment with IFN- α or exposure to pseudo-typed HIV-1 particles. This phenomenon can be attributed to two factors. First, CNP1 proteins maybe translated from CNP1 mRNA or alternatively from the second start codon of CNP2 mRNA. Second, CNP1 proteins can be generated from CNP2 proteins by cleaving N20aa. This dual process not only increases the quantity of CNP1 protein but also decreases the amount of CNP2 protein, thereby enhancing the host’s antiviral capacity.”

3. Q: Lines 160-162 and Fig. S2B: Observation that CNP2 phosphorylation mutants do not show predicted localization is never addressed/discussed. Possible reasons for these results should be discussed in the discussion.

A: We deeply appreciate your positive assessment of our work. Concerning the impact of CNP phosphorylation modifications on its mitochondrial localization (Lee, O’Neill et al, 2006), the current research is somewhat constrained. Nevertheless, our findings from laser confocal microscopy and mitochondrial fractionation experiments unequivocally demonstrate that phosphorylation at the S9/22 sites of CNP2 has no significant impact on its localization in the mitochondria. Additionally, predictions from the TargetP-2.0 database suggest a low likelihood

of the N20aa segment in CNP2 acting as a mitochondrial targeting signal (MTS). It is noteworthy that UniProt does not explicitly highlight the influence of CNP phosphorylation modifications on CNP localization (www.uniprot.org/uniprotkb/P09543/entry). We have chosen a succinct explanation in this context, with the anticipation that our research will incite further inquiry from scholars in the field.

4 & 5. Q: Lines 171-173 and Fig. 2F and 2G: Could GFP tagging cause protein's mislocalization in the cell? This possibility is not mentioned and should be either addressed in the results section or discussed in the discussion. Furthermore, in lines 174-176, it is mentioned that the N-terminal 20 amino acids of CNP2 were cleaved from GFP; if true, is this really a good experiment to do to determine the localization of the N-terminal 20 amino acids of CNP2? This should be discussed further and/or supported by additional experiments such as using an HA-tagged N-terminal 20 amino acids of CNP2, which can be stained with an anti-HA primary antibody followed by a secondary GFP conjugated antibody.

A: We really appreciated your question. In general, GFP did not affect protein's localization in cell, unless it affects the recognition of protein signal peptides or interferes with protein-protein interactions. In our system, we fused the N-terminal 20 amino acids (N20aa) of CNP2 to the N-terminus of GFP to validate its potential mitochondrial targeting signal function. The positive control COX8A-MTS-GFP showed clear localization in mitochondria. This approach has been well-established (J Cell Biol 2018; 217: 1369–1382).

However, since N20aa was cleaved from CNP¹⁻²⁰-GFP, CNP¹⁻³⁰-GFP and CNP¹⁻⁴⁰-GFP. GFP could not determine the localization of the N-terminal 20 amino acids of CNP2.

Fortunately, the aim of this experiment is to test: 1) whether N20aa function as MTS to direct CNP2 into the mitochondria; 2) whether N20aa is cleaved in the mitochondria. The answers to above question rely on the localization of GFP regardless the cleavage of N20aa. If N20aa functions as MTS, GFP signal would be observed in the mitochondria even after CNP2 N20aa is cleaved in the mitochondria. If N20aa is cleaved out the mitochondria, N20aa even has no chance to mediate CNP2 translocation. We found that GFP dispersedly

distributed in the cytoplasmic matrix of 293T cells expressing CNP2¹⁻²⁰-GFP, but not in the mitochondrial fraction. In addition, CNP1 without N20aa could translocate into the mitochondria, suggesting CNP1 inherently possesses mitochondrial targeting properties. All these data point to the notion that translocation of CNP into the mitochondria does not rely on N20aa.

We have also considered an experiment as you suggested to use a HA-tagged N-terminal 20 amino acids of CNP2. Unfortunately, if HA was tagged tag on the C-terminal of N20aa, N20aa would be also cleaved as CNP¹⁻²⁰-GFP; if HA was tagged tag on the N-terminal of N20aa, the function of N20aa as MTS (if it had) would be hindered, theoretically. Similar to HA-CNP¹⁻²⁰-GFP mutant, we found Flag-CNP2 mutant still localized to both the mitochondria and the cell membrane (Fig S3D and E).

According to your suggestion, we have made this aspect more clearly in the revised version of the manuscript (line 157-168), and hope this clarification addresses your concerns properly.

6. Q: Lines 231-232: Is the hypothesis of this statement/conclusion that tagging CNP2 at its N-terminus with a sfGFP11 prevents the cleavage of the N-terminal 20 amino acids, as Flag tag did, and so you can observe the non-cleaved sfGFP11-CNP2 at the cell membrane? This should be explicitly stated in this results section as it is critical to understanding this results section's conclusion.

A: According to your suggestion, we have performed experiments. We found that tagging sfGFP11 at CNP2 N-terminus prevents the cleavage of the N-terminal 20 amino acids. We also observed that the non-cleaved sfGFP11-CNP2 at the cell membrane. We have presented new experimental evidence (Fig S5) and revised the results section accordingly (line 221-224).

7. Q: General comment about the conclusion: A lot of it is focused on other, similar ISGs and their antiviral activity, which is fair and important to include as they share some similarities with

CNPs; however, the discussion should include/address multiple observations from this study, some of which I have pointed out in my comments (see Major Points 2, 3, and 4).

A: As we mentioned above, we have revised the sections of results and discussion. Your suggestions help us to improve the quality of our paper dramatically.

Minor Points

1. Q: Figure 1 label, line 11: "transfected with CNP2, ..."

Has been Corrected in the revised version of the manuscript (line 567-568)

2. Q: Line 193: The fact that post-translational processing of the N-terminal 20 amino acids of CNP2 may occur in the cytomatrix should be further elaborated on in the discussion; experiments needed to determine whether post-translational processing indeed occurs in the cytomatrix could be proposed.

A: In response to the reviewer's suggestion to provide more detailed information about the potential post-translational processing of the N-terminal 20 amino acids of CNP2 within the cytomatrix, we have carefully provided additional supporting evidence (Fig S3E, line 180-181). These findings have now been incorporated into the discussion as requested, specifically in the section indicated as line 288-304.

“Additionally, CNP2¹⁻²⁰-GFP, CNP2¹⁻³⁰-GFP, and CNP2¹⁻⁴⁰-GFP failed to target mitochondria, instead primarily localizing within the cytosolic fraction (Fig 2F). It is noteworthy that N20aa was cleaved from CNP2¹⁻²⁰-GFP, CNP2¹⁻³⁰-GFP, and CNP2¹⁻⁴⁰-GFP, rendering GFP an unreliable indicator of the localization of these mutants. Despite this limitation, our results counter the notion that N20 mediates CNP2 translocation and cleavage in the mitochondria. If N20aa functions as an MTS and is cleaved within the mitochondria, GFP signals should be observable in the mitochondria. Conversely, if N20aa is cleaved outside the mitochondria, it cannot facilitate CNP2 translocation. Confocal imaging experiments convincingly demonstrate that N20aa does not direct GFP signals to various cellular organelles, including mitochondria, endoplasmic reticulum, Golgi apparatus, cytoskeleton, and lysosomes. Intervention of cytoskeleton assembly and proteins transport from the endoplasmic reticulum to the Golgi apparatus did not affect processing of CNP2. Prevention the cleavage of N20aa through the

addition of the Flag tag, Flag-CNP2 retains localization to the cell membrane and mitochondria. Considering this evidence, it is highly probable that the cleavage of the N-terminal 20 amino acids of CNP2 occurs within the cytomatrix. Further investigations are warranted to elucidate the molecular mechanisms governing the post-translational processing of CNP2, with a specific emphasis on identifying the protease responsible for cleaving N20aa from CNP2.”

December 15, 2023

RE: Life Science Alliance Manuscript #LSA-2023-02188-TR

Prof. Hui Zeng
Beijing shijitan Hospital, Capital Medical University
Biomedical Innovation Center
No.10 Tie Yi Road, Yangfangdian, Haidian District
Beijing 100015
China

Dear Dr. Zeng,

Thank you for submitting your revised manuscript entitled "2',3'Cyclic nucleotide 3'phosphodiesterase 1 functional isoform antagonizes HIV-1 particles assembly". We would be happy to publish your paper in Life Science Alliance pending final revisions necessary to meet our formatting guidelines.

- please add your main, supplementary figure, and movie legends to the main manuscript text after the references section
- please add the Twitter handle of your host institute/organization as well as your own or/and one of the authors in our system
- Correct the name discrepancy in the presentation of the name of one of your co-authors. Please correct: Guo Li is in the system vs. Guoli Li is in the manuscript.
- please consult our manuscript preparation guidelines <https://www.life-science-alliance.org/manuscript-prep> and make sure your manuscript sections are in the correct order
- please use the [10 author names et al.] format in your references (i.e., limit the author names to the first 10)
- Correct figure S1, which has only one panel, so there is no need to label it as A. Please correct the figure, and its legend and callout in the manuscript text.
- On the title page, please use only the title, list of the authors and their affiliation, and corresponding authors' information.
- please add an Author Contributions section to your main manuscript text
- please add callouts for Figures S3C, F and S4C, E to your main manuscript text

Figure Checks:

- please add sizes next to all blots

A. FINAL FILES:

B. MANUSCRIPT ORGANIZATION AND FORMATTING:

Sincerely,

Reviewer #2 (Comments to the Authors (Required)):

In this manuscript, Liang et al. endeavour to elucidate the restriction of human immunodeficiency virus 1 (HIV-1) assembly by 2', 3'-cyclic nucleotide 3'-phosphodiesterase proteins (CNPs). CNPs are interferon stimulated genes (ISGs) that have been previously shown to suppress viral replication of HIV-1, simian immunodeficiency virus (SIV), and severe acute respiratory syndrome coronavirus 2 (SARS-CoV-2). Two isoforms of CNPs have been identified, CNP1 and CNP2. CNP1 and CNP2 are encoded by a single gene with alternative transcription initiation from respective promoters. Compared to CNP1 proteins, CNP2 proteins have an extra N-terminal 20 amino acids, which have been recognized as mitochondrial targeting sequence (MTS). While CNP2 has been previously shown to block HIV-1 particle assembly, it is unclear whether CNP1 has the same antiviral activity. The authors address this question by using a pseudovirus system and doxycycline (dox) inducible expression system. To elucidate the localization of CNP proteins in the cell and the site of their interaction with HIV-1 Gag protein, they performed a high-resolution 3D reconstruction confocal imaging. The authors conclude that 1) CNP1, a truncated form of CNPs, has anti-HIV-1 activity, 2) post-translational processing of N-terminal 20 amino acids of CNP2 occurs in the cytoplasmic matrix and is prerequisite for its antiviral activity, 3) CNP1 proteins interact with HIV-1 Gag protein on the cell membrane, and 4) N-terminal 20 amino acids of CNP2 inhibit CNP2 interaction with Gag protein on the cell membrane. These conclusions are based on four key pieces of data:

- 1) Co-transfection of either CNP1 or CNP2 expression plasmid with HIV-1-based lentiviral pseudotyped particle assembly vectors in 293T cells results in more than 1,000-fold reduction of luciferase activity and reduced p24 levels in the culture media/increase of Pr55Gag precursors in transfected 293T cells. - Suggesting that CNP1, a truncated form of CNP2, has anti-HIV-1 activity.
- 2) A) To inhibit the cleavage of N-terminal 20 amino acids of CNP2, they generated Flag-CNP2. To eliminate alternative CNP1 proteins' translation from CNP2 mRNA, they generated CNP2-M21L mutant. Co-transfection of either Flag-CNP2 or CNP2-M21L expression plasmid with HIV-1-based lentiviral pseudotyped particle assembly vectors in 293T cells exhibits reduced levels of CNP1 proteins and enhanced levels of CNP2. Compared to wild-type CNP2, Flag-CNP2 shows reduced anti-HIV-1 activity. - Suggesting that N-terminal 20 amino acids of CNP2 post-translational processing to form CNP1 is a prerequisite for anti-HIV-1

activity.

2) B) To determine where post-translational processing of N-terminal 20 amino acids of CNP2 occurs, they established stably transduced 293T cell lines expressing CNP1 and CNP2 based on the pLVX-TetOne lentiviral vector. In both CNP1 and CNP2 stably transduced 293T cells, Western blot analysis revealed the presence of CNP1 proteins in the mitochondrial fractions which is confirmed with confocal imaging showing that CNP1 co-localises with the mitochondrial marker Tom20.

2) C) To further support the observations above, Liang et al. constructed CNP2 mutants lacking either 30 or 40 N-terminal amino acids; both mutant proteins also translocated into the mitochondria. - Suggesting that CNP1 inherently possesses mitochondrial targeting properties. They also generated stably transduced 293T cell lines with a CNP2-S9/22A mutant construct in which phosphorylation of Ser9 and Ser22 was blocked, or with a CNP2-S9/22D mutant to mimic CNP2 phosphorylation at Ser9 and Ser22. Western blot analysis revealed the presence of CNP2 and CNP1 proteins in the mitochondrial fraction in 293T cells stably transduced with these mutants; this was confirmed with confocal imaging which showed the colocalisation of CNP signals with Tom20 signals. As per reviewers' suggestions, Liang et al. sought an alternative approach to assess whether the N-terminal 20 amino acids of CNP2 functioned as a MTS: they engineered CNP21-20-GFP, CNP21-30-GFP, and CNP21-40-GFP mutants encoding fusion proteins of N-terminal 20, 30, or 40 amino acids of CNP2 fused with GFP, and a COX8A-MTS-GFP mutant encoding a fusion protein of human COX8A MTS and GFP as a positive control. Western blot analysis and confocal imaging confirmed that COX8A MTS directed GFP translocation into the mitochondria while GFP signals from CNP21-20-GFP, CNP21-30-GFP, and CNP21-40-GFP were found in the cytosolic fractions. - Suggesting that the N-terminal 20 amino acids of CNP2 do not function as MTS to mediate GFP translocation into the mitochondria.

3) Generation of 293T cells with a stable expression of HIV-1 Gag proteins and dox-induced expression of CNP variants; upon CNP1 induction by dox, CNP1 co-localized with HIV-1 Gag proteins and cell membrane protein CD81 but not with mitochondria protein Tom 20. - Suggesting that CNP1 interacts with HIV-1 Gag on the cell membrane.

4) Co-expression of either split super-folder GFP11-CNP1 (sfGFP11-CNP1) or sfGFP11-CNP2 with Gag containing a split sfGFP10 (Gag-sfGFP10) in 293T cells shows approximately 30% decrease in GFP level in sfGFP11-CNP2 compared to sfGFP11-CNP1. - Suggesting that the N-terminal 20 amino acids of CNP2 inhibit CNP2 interaction with Gag protein on the cell membrane.

Liang et al. addressed all reviewers' comments and revised their MS accordingly. The MS has improved both in how it reads as well as how the data are presented. There is still room for some language improvement, otherwise, I am happy with the acceptance of this MS w/o further revisions.

December 21, 2023

RE: Life Science Alliance Manuscript #LSA-2023-02188-TRR

Prof. Hui Zeng
Beijing shijitan Hospital, Capital Medical University
Biomedical Innovation Center
No.10 Tie Yi Road, Yangfangdian, Haidian District
Beijing 100015
China

Dear Dr. Zeng,

Thank you for submitting your Research Article entitled "2',3'Cyclic nucleotide 3'phosphodiesterase 1 functional isoform antagonizes HIV-1 particles assembly". It is a pleasure to let you know that your manuscript is now accepted for publication in Life Science Alliance. Congratulations on this interesting work.

DISTRIBUTION OF MATERIALS:

Again, congratulations on a very nice paper. I hope you found the review process to be constructive and are pleased with how the manuscript was handled editorially. We look forward to future exciting submissions from your lab.

Sincerely,
